# Mucositis and Infection in Hematology Patients

**DOI:** 10.3390/ijms24119592

**Published:** 2023-05-31

**Authors:** Nicole M. A. Blijlevens, Charlotte E. M. de Mooij

**Affiliations:** Department of Haematology, Radboud University Medical Center, P.O. Box 9101, 6500 HB Nijmegen, The Netherlands; c.demooij@radboudumc.nl

**Keywords:** mucositis, inflammation, bacteremia, infection, chemotherapy, neutropenia

## Abstract

Survival in patients with hematological malignancies has improved over the years, both due to major developments in anticancer treatment, as well as in supportive care. Nevertheless, important and debilitating complications of intensive treatment regimens still frequently occur, including mucositis, fever and bloodstream infections. Exploring potential interacting mechanisms and directed therapies to counteract mucosal barrier injury is of the utmost importance if we are to continue to improve care for this increasingly growing patient population. In this perspective, I highlight recent advances in our understanding of the relation of mucositis and infection.

## 1. Introduction

For a clinician, treating patients with hematological cancers and dealing with mucositis and infection is similar to Odysseus’ choice between Scylla and Charybdis, i.e., the necessity to choose the lesser of two evils (Homer, Odyssey, Book XII, vs. 234–260). In the case of mucositis and infection, this choice has been almost always the disaster of infection because we have drugs to treat or prevent infections. Infections caused by microorganisms are a life-threatening complication in patients treated for their cancer with chemotherapy and other modalities, especially patients in hematology who are immunocompromised by their malignant disease interfering with normal immunological defense. The updated ‘European Hematology Association (EHA) Research Roadmap: Infections in Hematology’ published in 2021 highlighted the urgent clinical and scientific needs considering infection in hematology [1]. Undeniably, the use of anti-infective drugs significantly reduced chemotherapy and transplant-related morbidity and mortality. Despite these advances, infections remain important causes of non-relapse mortality. It has become clear that a deeper understanding of the gut–immune axis in the context of hematological diseases may revolutionize future treatment strategies, especially concerning infections in neutropenic patients [1]. Recent (inter)national guidelines, for instance ‘Antimicrobial Prophylaxis for Adult Patients With Cancer-Related Immunosuppression: ASCO and IDSA Clinical Practice Guideline Update’ [2], the Dutch ‘Working Party on Antibiotic Policy (SWAB) Recommendations for the Diagnosis and Management of Febrile Neutropenia in Patients with Cancer’ [3], or the ‘Guideline for the Management of Fever and Neutropenia in Pediatric Patients With Cancer and Hematopoietic Cell Transplantation Recipients: 2023 Update’ [4] lack focus on the clinical impact of the gut–immune axis and the role of mucositis in relation to preventing or treating infections. On the other hand, the ‘MASCC/ISOO clinical practice guidelines for the management of mucositis secondary to cancer therapy’ [5] and the ‘MASCC/ISOO clinical practice guidelines for the management of mucositis: sub-analysis of current interventions for the management of oral mucositis in pediatric cancer patients’ [6] focus solely on mucositis and not on the infectious complications of that damaged gut–immune axis in neutropenic patients with fever. Fever is often the sole sign of a possible infection in an immunocompromised patient because of the lack of neutrophils that are normally responsible for inducing the classical symptoms of a host defense to infections, for instance redness (rubor), heat (calor), swelling (tumor), and pain (dolor). The adage in daily clinical practice therefore has always been it is an infection until proven otherwise. As the mucosal membranes are packed with microorganisms, it should not be considered strange to consider mucositis presenting itself with classical symptoms of local inflammation to be an infectious complication of the administrated chemotherapy too and consequently managed as such. Moreover, the lack of proper anti-mucositis treatments left us telling ourselves we had no better remedies. The incidence of oral mucositis (OM) is underreported by clinicians and ranges from about 15% in chemotherapy-treated cancer patients to more than 90% in total body irradiation (TBI) included myeloablative-conditioned hematopoietic cell transplant (HCT) recipients [7]. Severe OM (WHO grade 3–4) was reported in 44% after high-dose melphalan or BEAM-conditioned autologous HCT in one of the few multicenter prospective studies with special attention on well-trained examinations of OM [8]. Incidence rates of gastrointestinal mucositis are even more difficult to establish because proper scoring tools are lacking and related symptoms such as nausea or diarrhea are multifactorial and influenced by treating agents and administrated supportive care drugs. In this article, we want to try separating mucositis and infection both on a mechanistic level and on the level of clinical management and suggest new therapeutic options to support clinicians sailing between Scylla and Charybdis.

## 2. Is Mucositis an Infectious Disease?

Mucositis of the mouth is clinically characterized by pain, erythema, oedema and ulcerations and mucositis of the alimentary tract by pain, bowel cramps, nausea, vomiting and diarrhea. This both results in anorexia, weight loss and reduced quality of life, delay of effective anticancer treatments and sometimes even ICU transfer and death [7]. The pathogenesis of mucositis is complex, dynamic and intricate, and the precise role of microorganisms is still the subject of debate [9]. Injury to the mucosa induced by chemotherapy or irradiation is definitely the most substantial and earliest breach in the host defenses against microorganisms. However, other components of the physical and functional barrier, i.e., mucus layer, neuroendocrine feedback signaling, the immune system itself and the vascular system are injured as well by the chemotherapy or irradiation. As mentioned, the process of mucositis is more complex than just the direct effect of cytotoxic therapy on cells with a high mitotic index, such as epithelial cells of mouth and gastrointestinal tract resulting in apoptosis and a maturation blockade for many days. The postulation of Sonis [10] about the pathobiology of oral mucositis involving five sequential and partially overlapping phases has never really been challenged. The initiation phase involves free radical generation, induction of apoptotic cell death induced by both DNA and non-DNA damage and activation of the innate immune response [11]. In addition, generated (radical oxygen species) ROS lead to lipid peroxidation, sphingomyelinase activation, and the hydrolysis of membrane sphingomyelin to yield ceramide. Although ceramide is considered a proapoptotic molecule, its accumulation is a signal for increased membrane permeability and ultimately break of the epithelial cells. The following phase is crucial during which the master transcription factor, nuclear factor kappa B (NF-κB), leads to the upregulation of several genes, resulting in the production of the proinflammatory cytokines (tumor necrosis factor-α [TNF-α], IL-1β, IL-4, IL-6 and IL-18) and also cyclooxygenase 2 and cell adhesion molecules. These proinflammatory cytokines turn on a cascade of reinforced signal amplification. Simultaneously connective tissue fibrinolysis and the stimulation of tissue-damaging matrix metalloproteinases further damage the extracellular matrix. The net result of apoptosis, maturation blockade, necrosis and inflammation is the ulceration phase, in which microorganisms and microbe-associated molecular patterns such as peptidoglycans and lipopolysaccharides can translocate the damaged mucosal barrier more easily and microorganisms can cause systemic infections, especially in the absence of circulating neutrophils and monocytes. Damage-associated molecular patterns, including endogenous ligands, the alarmins, high-mobility box group-1, ATP, and heat shock proteins released as a result of mucosal damage, in addition to (translocated) microorganisms and microbe-associated molecular patterns, are able to further activate tissue macrophages to produce even more proinflammatory cytokines, all contributing to profound inflammation and mucosal tissue damage [11,12,13]. Cross-talk between all of the active elements of the physical and functional gut barrier is intense and multidirectional. Inflammation not only mitigates infections but initiates tissue repair. The same pro-inflammatory cytokines IL-6 and TNF-α promote regeneration of the intestinal mucosa [14]. Epithelial stem cells are capable of repopulating the mucosa in the healing phase and the return of neutrophils will clear up damaged cells and infected tissue [15]. As a result of the ulcerative surfaces, the oral flora undergoes a shift towards more pathogenic bacteria as will be discussed later in more detail. Although a few studies have established changes in microbiome composition or relative proportions, it is unclear how these influence the course of OM, as all five stages can coexist to a variable degree across the oral mucosa [16]. To hypothesize, the oral microbiome may play a role in exacerbating or protecting against mucosal injury during the five stages of mucositis. Such an impact may be determined by changes in the microbial expressome and interactome, species adhesion properties, or colonization activities, which are concomitant to or followed by inflammation and ulceration [15]. In the absence of microbes, tissue damage and cell death still evoke sterile inflammation [17]. To summarize, there is no evidence suggesting that microorganisms by themselves induce chemotherapy-induced mucositis although microbial colonization might be an important factor in the propagation or amelioration of the inflammatory response seen in mucositis.

## 3. Does Antimicrobial Therapy Ameliorate Mucositis?

In a recent review of Colella et al. [18], evaluating effective methods tested in randomized controlled trials (RCTs) in prevention or reduction of incidence or severity of oral mucositis, Collela et al. found that anti-inflammatory medications for instance benzydamine was effective in the prevention of radiotherapy- and chemoradiotherapy-induced oral mucositis. The results varied when they evaluated RCTs testing antimicrobial agents. Chlorhexidine was the main antimicrobial agent studied for those patients undergoing chemotherapy and chemoradiotherapy with three randomized clinical studies of Dodd et al., Epstein et.al., and Diaz-Sanchez et.al. reporting no statistically significant differences between the different groups [19,20,21]. Two small studies of Cheng et al. and one large study of Sorensen [22,23,24] reported statistically significant differences favoring the chlorhexidine group, but none of the studies actually attempted to recover the microorganisms by culture to determine or evaluate the effect of chlorhexidine on them, nor their effects on mucositis. Iseganan, an analog of protegrin-1, a naturally occurring peptide with broad-spectrum microbicidal activity was tested in a randomized, double-blind, placebo-controlled study in hemato-oncology patients. Among all 323 (iseganan/placebo 163/160) patients, analyzed according to randomization assignment, 43% and 33% of iseganan and placebo patients, respectively, did not develop ulcerative oral mucositis (*p* = 0.067). On an 11-point scale, iseganan patients experienced less mouth pain (3.0 and 3.8 [*p* = 0.041]), throat pain (3.8 and 4.6 [*p* = 0.048]), and difficulty swallowing (3.9 and 4.7 [*p* = 0.074]), compared to placebo patients. On the 5-point NCI CTC scale, iseganan patients experienced lower mucositis scores (1.6 and 2.0 [*p* = 0.013]). Previous studies have demonstrated that iseganan produces rapid microbicidal activity in saliva, has a broad spectrum of antimicrobial activity against Gram-positive and Gram-negative bacteria and fungi, but over 80% of patients in both arms (iseganan/placebo) received antimicrobial prophylaxis (drugs: antibacterial 84/85 (%), antifungal 72/69 (%), antiviral 60/53 (%), antibacterial and antifungal 79/72 (%), antibacterial, antifungal and antiviral 48/47 (%)), which is the standard of care in these kind of HCT recipients [25]. Gram-positive microorganisms such as Staphylococcus species and Streptococcus species are the most isolated bacteria in the period of clinical oral mucositis. Azithromycin, a macrolide antibiotic with anti-inflammatory properties, modulates the inflammatory response by accumulation in the polymorphonuclear leukocytes and macrophages and exerts antimicrobial effects against staphylococci and streptococci and therefore could be effective in the prevention and treatment of oral mucositis. Parkhideh et al. performed a single-blinded, randomized, controlled clinical trial in HCT recipients. Azithromycin in the concentration of 200 mg/5 mL was used twice daily from the first day of chemotherapy until engraftment day or mucositis resolution. Out of 88 enrolled patients, 18 were excluded due to a prolonged neutropenia phase, death, variation of patients’ health status, gastrointestinal complications following transplantation, and lack of cooperation. Azithromycin use was associated with delayed onset (7.5  ±  1.4 vs. 5.3  ±  2.2, *p*  =  0.015) of oral mucositis and shorter duration of mucositis (5.1  ±  1.3 days versus 8.8  ±  4.1 days in control patients, *p*  =  0.045). However, there was no significant difference in the maximum grade of oral mucositis (*p*  =  0.157) or average daily grade of oral mucositis (*p*  =  0.298), nor in days of fever or average use of antibiotics. Two patients in the intervention and seven patients in the control group were found to have positive blood cultures (*p*  =  0.047), but details regarding bacterial species were not reported. Besides azithromycin, patients in both groups received sodium chloride 0.9% and chlorhexidine mouthwash 5 mL three times daily to reduce the risk of oral infections [26]. In another study, 132 patients were randomized to use normal saline (*n* = 65) or povidone-iodine diluted 1:100 (*n* = 67) mouthwashes for OM prophylaxis after high-dose chemotherapy comprising BEAM or HD-L-PAM followed by autologous peripheral HCT. No significant difference was found between the groups in respect of OM characteristics, fever of unknown origin (FUO) and other infections [27]. Hence, the data supporting the use of antimicrobials for the prevention of OM are at best inconclusive and it seems that we have not improved that much since the pivotal review of Donnelly almost 20 years ago [28]. A better understanding of the pathophysiological role of microorganisms in the different phases of mucositis is still very much needed. Manipulation of the microbiota offers a novel, intriguing and challenging target.

## 4. Might Insights of Microbiome Research Provide More Clinically Relevant Answers with Respect to Mucositis?

### 4.1. Oral Dysbiosis

Microbial communities exist on every mucosal surface in the human body, and each body site within a person has a unique ecology. Human-resident microorganisms encode an estimated 2 million to 20 million genes, whereas the human genome encodes an estimated 20,000 to 25,000; therefore, the microbiome represents up to 99.9% of the genetic capacity in the human body. Microbial communities act as a dynamic component of the body [29]. The oral cavity harbors a complex ecosystem of bacteria, fungi, viruses, archaea, and protozoa. About 1000 different microbial species can be found in the oral cavity. Much more is known about the bacteria in the oral cavity than about the other oral microorganisms. The genus Streptococcus is most abundant in the oral cavity, while different niches additionally contain other genera, such as Veillonella, Prevotella, and Fusobacterium, depending on the substrate (for instance enamel, (un)keratinized mucosa, or saliva) and site in the oral cavity (for instance supra- or subgingival) [16]. The vast majority of the oral microorganisms live in biofilms and previous work support the concept of a core microbiome in health [30]. A recent systematic review following the PRISMA protocol selected—out of 166 obtained articles—only 5 articles that met eligibility criteria to answer the 4 questions of (1) whether patients diagnosed with cancer, who are candidates for receiving systemic antineoplastics (P = Patients), and (2) undergo oral microbiome determinations (I = Intervention), (3) before and after systemic antineoplastics administration (C = Comparison), and (4) analyzed changes in the oral microbiome composition (O = Outcome) [31]. Acute myeloid leukaemia (AML) was the most frequent type of cancer (40%) among the participants. Only one of the studies included a control group of healthy subjects. Heterogeneity in the protocols and approaches of the included studies hindered a detailed comparison of the outcomes. A decrease in bacterial alpha diversity is often associated with oral mucositis. However, from literature, there is no clear association between specific bacterial species and oral mucositis. Differences between studies existed in bacterial strains studied, study population, collection time, sampling methods, and scoring methods for OM. Therefore, it is not possible to draw detailed conclusions from all of these studies [15]. In our prospective, two-center study (Ethics Committee number NL52117.018.15), we investigated the dynamics of microbial changes in relation to the development of ulcerative oral mucositis in fifty-one patients with multiple myeloma homogenously treated with high-dose melphalan followed by autologous hematopoietic stem cell transplant (AHCT) [31]. Twenty patients (39%) developed ulcerative OM (WHO scale). The oral microbiome determined by using 16S rRNA amplicon sequencing changed significantly after AHCT and returned to pre-AHCT composition after three months. Changes in microbial diversity and similarity were more pronounced and rapid in patients who developed ulcerative OM compared to patients who did not. Already before AHCT, samples from non-ulcerative OM patients contained a higher proportion of reads classified as *Actinomyces graevenitzii* and *Streptococcus constellatus*, while samples from ulcerative OM patients showed a higher proportion of reads classified as genus *Veillonella*, *Enterococcus faecalis*, *Streptococcus* spp., *Staphylococcus* spp., *Fusobacterium* spp., *Prevotella oris* and *Prevotella veroralis* suggesting microbially driven risk factors. Indicating that patients who did not develop ulcerative OM had a more resilient microbial ecosystem albeit there were differences with respect to antimicrobial prophylaxis. Samples with high fungal load by qPCR (>0.1%) had a significantly different microbial profile from samples without fungi. Bruno et al. [32] generated a machine learning-based bacterial signature that uses pre-treatment microbial profiles from 30 allogeneic HCT patients to predict whether a patient will develop OM during treatment. A total of 13 patients (43%) developed ulcerative OM. Relative abundance of *Porphyromonas* spp. at preconditioning was positively correlated with ulcerative OM grade (Spearman ρ = 0.61, *p* = 0.028) and higher relative abundance of *Lactobacillus* spp. at ulcerative OM onset was associated with shortened ulcerative OM duration (median time: 6 vs. 10 days, *p* = 0.032). With respect to healing of oral ulcers it is clear that several oral bacteria such as *P. gingivalis*, *Prevotella nigrescens*, *Streptococcus mitis*, *Tannerella forsythia*, and *P. intermedia* or their secreted molecules are capable of inhibiting the migration of oral epithelial cells in vitro [33].

### 4.2. Gut Dysbiosis

The relationship between epithelial integrity and the microbiome is more extensively studied in the intestine. The human gut contains up to 100 trillion bacterial cells that belong to as many as thousand different species [34]. The gut microbiome mainly consists of commensal microbes that exhibit a symbiotic relationship with their host, modulate nutrient metabolism and absorption, influence intestinal development and function and shape the gastrointestinal immune landscape [35,36]. The human intestine is normally in a state of low-grade inflammation to protect itself from infection being responsible as mentioned before for both pathogen killing and tissue repair processes. The microbiome mainly dictates the amplitude of the inflammatory response and its outputs [14]. Intensive chemotherapy [37,38,39] and HCT [40,41] in general, are associated with oral and gut microbial dysbiosis (lower microbial diversity and outgrowth of opportunistic pathogens). Microbial dysbiosis is considered to play a role in severe (post-transplant) complications such as bloodstream infections or bacteremia [37,42] and pulmonary infections [43], graft-versus-host disease [44,45], perturbed immune reconstitution [46], even relapse (higher abundance of mostly Eubacterium limosum) [47] and non-relapse mortality [40]. Not unsurprisingly, antibiotics have been the focus of most studies investigating microbial changes in HCT recipients given the high use of broad-spectrum, gut decontaminating antimicrobials in this immunocompromised patient population experiencing (recurrent) episodes of neutropenic fever [48,49,50]. Antibiotics have been attributed to more extensive GvHD and worse HCT outcomes [51,52]. Chemotherapy/radiation and early alloreactivity can induce changes in microbiome composition [44] and microbiome manipulations can attenuate or exacerbate the effects of chemotherapy, radiation, and alloreactivity [53,54,55]. 

Using high-throughput DNA-sequencing analysis of stool samples of non-Hodgkin’s lymphoma patients who received the same myeloablative conditioning regimen and no other concomitant therapy such as antibiotics identified not only significant decreases in abundances of Firmicutes and Actinobacteria and significant increases in abundances of Proteobacteria compared to samples collected before chemotherapy (complementary to other reports), but also a reduced capacity for metabolism of nucleotide, energy, cofactors and vitamins, and an increased capacity for glycan metabolism, signal transduction and xenobiotics biodegradation [56]. These changes in metabolic function may indicate attempts of the intestinal microbiota to resist oxidative stress induced by intestinal inflammation after cytotoxic therapy. Several taxa that we found decreased after chemotherapy, such as Faecalibacterium, Ruminococcus, Coprococcus, Dorea, Lachnospira, Roseburia and Clostridium, are well-known to diminish inflammation by modulation of the NFκB pathway [57]. Decreases of Bifidobacterium following chemotherapy also have the ability to inhibit inflammation in intestinal epithelial cells through attenuation of TNF-α and lipopolysaccharide (LPS)-induced inflammatory responses [58] while depleting taxa of butyrate-producing bacteria, which results in reduced production of short chain fatty acids (SCFAs) necessary to maintain homeostasis in the colonic mucosa and inhibit inflammatory response and is associated with high risk lethal GI-GvHD [59]. Other bacteria increased after chemotherapy such as Citrobacter have the ability to activate NFκB and therefore may increase inflammatory responses. The depletion of Faecalibacterium, Ruminococcus, Coprococcus, Dorea, Lachnospira, Roseburia, Clostridium and Bifidobacterium after chemotherapy is associated with increased intestinal permeability through NFκB and TNFα inhibition [31,32]. A decrease in Bifidobacterium after chemotherapy therefore may be detrimental to the maintenance of efficient barrier function. A decrease in butyrate-producing bacteria after chemotherapy also reduces intestinal permeability and leads to barrier dysfunction. Furthermore, previous studies reported that LPS-induced inflammation increased intestinal permeability through TLR-4-dependent up-regulation of CD14 membrane expression. Butyrate-producing bacteria play a role in the composition of the mucus layer, as butyrate has the ability to increase mucin synthesis via MUC2. The mucus layer may also be compromised by specific pathogens, such as Enterobacteriaceae, which can form biofilms on the epithelial surface that alter the mucus layer [60]. Citrobacter may participate in the degradation of the mucus barrier, using mucinases or glycosidases to digest mucin. Others reported increases in the pathogenic *Escherichia coli* and *Enterococcus* spp., which were shown to be associated with concomitant histologic impairment of the gut and an emerging role of bacterial proteases on the mucosal barrier function [61,62,63].

### 4.3. Bidirectional Relationship between Dysbiosis and Mucositis

The clinical picture in case of the role of dysbiosis in onset, severity and duration of oral and gut mucositis is really complex as the simultaneity of the microbial changes (induced by chemotherapy and/or antimicrobial therapy) and mucositis makes a cause-and-effect analysis using clinical case–cohort data inherently difficult. In animal experiments, at least, effects in both directions have been observed. It is critical to consider the other factors that likely influence the gut microbiota in HCT recipients, including the use of high dose chemotherapy itself, mucosal inflammation and diet; each of which have been associated with microbial injury [64,65,66]. In preclinical models of chemotherapy-induced mucositis without alloreactivity, a decrease in both the number and diversity of bacteria was seen, which was shown to be correlated with the presence of diarrhea and reduced villus length. We developed a rat model for treatment with melphalan, in which important clinical parameters of high-dose melphalan (HDM)-induced toxicity are represented. Melphalan in a dose of 5 mg/kg induced marked intestinal mucositis, reflected by a rapid decrease in citrulline, a biomarker of total enterocyte mass. Citrulline-based assessment of intestinal damage has been shown to be objective, reproducible, specific and reliable in detecting intestinal mucositis in the setting of intensive cytotoxic therapy [67]. Melphalan caused severe histopathological injury in the small and large intestine. Weight loss and diarrhea caused by melphalan were dose-dependent. The citrulline course in rats treated with melphalan 5 mg/kg was remarkably similar to that seen in patients, apart from occurring over a shorter period of time. Other important clinical correlates, such as increased body temperature and neutropenia, were also observed in this animal model. Melphalan induced disruption of the microbiota characterized by pathogen expansion and SCFA deficits. Finally, melphalan-induced injury was shown to result in bile acid malabsorption and a decrease in primary to secondary bile acid ratios in plasma [66], and metabolites that interfere with immunological pathways. Rashidi et al. [68] proposed a model in which the commensal microbe Blautia protects against neutropenic fever (NF) in a matched case–control analysis of allogeneic HCT recipientsand patients with acute leukemia treated with chemotherapy. A greater abundance of *Blautia* spp. predicts higher citrulline levels, but can be easily reversed by suggesting that the integrity of the intestinal barrier (indicated by low citrulline levels) might dictate a lower *Blautia* spp. abundance. The class of anaerobes is normally dependent on an intact mucus barrier that is damaged by MBI, and therefore changes in *Blautia* spp. may simply reflect preceding changes in epithelial integrity. Changes in microbiota usually occur after the use of (prophylactic) antibiotics because of NF and after the onset of MBI [69]. Furthermore, temporal changes in diversity mirroring changes in citrulline. Loss of *Blautia* spp. may reflect the extent of microbial-toxicity induced by chemotherapy such as citrulline reflects gut MBI and that the chemotherapy-induced toxicity itself results in fever (“febrile mucositis”) [70]. Interestingly, recent studies reported the intensity of chemotherapeutic regimens to be associated to the extent of dysbiosis [40,71]. The report by Shouval et al. [71] found an association between the extent of diversity loss and the intensity of conditioning that was independent of antibiotics. Through longitudinal analyses of 210 stool samples, with matched blood and clinical metadata, we provide the first evidence to indicate that intestinal mucositis initiates microbial dysbiosis in a homogenous cohort of AHCT recipients [72]. This was by accompanied associated changes in the microbial metabolome, with rapid and persistent loss of short chain fatty acids. Importantly, we show that intestinal mucositis, determined by plasma citrulline, predicts the fecal microbiota in AHCT recipients, and in fact precedes it by 9 days. This provides a unique opportunity to predict patients at risk of microbial-mediated complications (e.g., bloodstream infection), with the time to intervene proactively. Furthermore, these data redirect attempts to support the gut microbiota indirectly by controlling intestinal mucositis. In conclusion, intestinal mucositis might be the most significant factor influencing the composition of the gut microbiota in AHCT recipients. Strategies aiming to promote microbial stability or resilience must be designed to address the influence of mucositis on microbial composition. While this finding indicates that conditioning makes an independent contribution to diversity loss, it does not mean antibiotics do not make an (additional) independent contribution (Figure 1). 

## 5. Are Infections and Fever during Neutropenia Caused by Mucositis?

The occurrence of bloodstream infections (bacteremia) during neutropenia increases the risk of admission to the intensive care unit as well as non-relapse mortality [74,75]. Infections are associated with graft-versus-host disease (GvHD) and due to the use of antibiotics, there is an increased risk of antimicrobial resistance as well as disturbance of the microbiota (dysbiosis) as mentioned before. The paradigm of ‘febrile neutropenia’ was the foundation for the strategy of prompt starting of antimicrobial therapy as soon as fever occurs during neutropenia, in order to reduce the risk of sepsis and death [76]. Empirical broad-spectrum antimicrobial therapy is still the standard of care and has resulted in improved survival in cancer patients [77]. Antimicrobial prophylaxis is commonly given to patients receiving intensive cytotoxic therapy who are expected to suffer from severe neutropenia [78]. However, this practice is increasingly being questioned due to emerging antimicrobial resistance, increased risk for infections with *Clostridioides difficile*, and detrimental effects on the microbiota [79]. Recent studies incorporating microbiome sampling clearly showed that loss of diversity and intestinal domination of opportunistic bacteria precede bacteraemia by the same pathogen in HCT and leukemia patients [38,39,42]. It took many years, however, to acknowledge that MBI and subsequent bacterial translocation from the gut is a major cause of BSI, resulting in the introduction of ‘MBI laboratory-confirmed BSI’ (MBI-LCBI) by the CDC in 2013, thereby differentiating between central line-associated BSI (CLABSI) and MBI-LCBI [80,81]. Subsequently, it was shown that, particularly in hemato-oncology patients, MBI-LCBI was a frequently occurring complication [82,83]. It became recognized that MBI-LCBI in HCT patients is associated with significant morbidity and mortality, and by extension leads to increased use of health care resources [84,85]. We performed a retrospective analysis of 628 episodes of CVC use in 508 patients managed with a CVC for remission induction chemotherapy (RI, ‘3 + 7’), or conditioning for autologous and allogeneic HSCT (MAC and NMA/RIC). We showed that, in general, bacteremia occurs frequently, although BSI risk differs significantly between treatment groups. We analyzed the grade of intestinal mucositis of each treatment regimen, using the biomarker citrulline. For all treatment regimens, we calculated the area under the curve of hypocitrullinemia <10 µmol/L (AUC_citrulline_), as well as the duration of hypocitrullinemia. The grade of intestinal mucositis strongly correlated with the occurrence of bacteremia. In NMA/RIC, the mean duration of hypocitrullinemia was short, with LCBI and MBI-LCBI occurring infrequently. In contrast, in autologous, RI and MAC allogeneic regimens, LCBI and MBI-LCBI occurred frequently, and the mean duration of hypocitrullinemia was significantly higher. Therefore, we concluded that citrulline, and specifically duration of hypocitrullinemia below 10 µmol/L, can be used to grade the bacteremia risk of chemotherapy and conditioning regimens would enable better tuning of the need for antibiotic treatment [86]. BSI occurs on average 12–14 days following intensive chemotherapy [87]. This coincides with the peak of permeability of the gut, as well as the development of severe intestinal MBI as determined by severe hypocitrullinemia (<10 µmol/L). The association with neutropenia appears to be much weaker because the incidence of bacteremia in non-myeloablative conditioning regimens is lower although neutropenia lasts longer when compared to myeloablative regimens [88]. The introduction of hypomethylating therapy in AML and myelodysplastic syndrome induces no mucositis nor hypocitrullinaemia [86] and very few bacteremias [89]. As mentioned before, despite significant neutropenia, non-myeloablative regimens are associated with very little intestinal mucositis as well as limited inflammation. This is consistent with preclinical studies in which the release of pro-inflammatory cytokines was associated with evolving gut mucosal damage and preceded microbial translocation [13]. This might also explain why giving antibiotics preemptively to reduce the occurrence of fever usually fails, as these antibiotics have little or no impact on mucositis evolution [90]. These findings suggest that the severity of mucosal damage in chemotherapy treated patients better defines the risk period of bacteremia than neutropenia. Despite shortening the duration of neutropenia with the use of hematopoietic growth factors, such as granulocyte colony-stimulating factor, this only marginally reduced infection-related mortality [91]. The diagnostic microbiological work-up in case of neutropenic fever only accounts for 20–40% positive blood cultures, but this number depends on the intensity of sampling at the onset of febrile neutropenia. Even before FN occurs blood cultures became positive in the advent of mucositis [92]. Fever remains unexplained in 30–40% of neutropenic patients, as there is no evidence of a clinically or microbiologically defined infection, and persists for 4–5 days or even longer in approximately 30% of patients despite the use of broad-spectrum antimicrobial therapy and advanced diagnostics [93,94,95,96]. Of course in case of a systemic inflammatory response not related to micro-organisms, antimicrobial therapy will fail to defervescence. In a study of leukemia-treated patients using serum flagellin as a culture-independent surrogate, it was again confirmed that specific changes in the indigenous gut microbiota preceded bacteremia and that serum flagellin was significantly higher in the group of patients with NF [97]. They proposed an alternative mechanism whereby expansion of certain bacteria (not necessarily pathogens) alters gut barrier and/or microbiota facilitating the translocation of pathogens. However, the dysbiosis itself may be altered by the evolving mucositis in a bidirectional manner. A recent paper re-invented the use of lactulose/rhamnose dual sugar absorption test in HCT recipients and reported that higher L/R ratios (increased gut permeability) were associated with lower microbiome diversity, loss of anaerobic organisms and a higher plasma lipopolysaccharide binding protein [98]. Whether inflammatory cytokines and/or DAMPS and/or MAMPs modulate intestinal permeability, that final demise in the barrier is necessary before any bacterial translocation can result in bacteremia during neutropenia [9,99]. Microbe-associated molecular patterns (MAMPs) and danger-associated molecular patterns (DAMPs) sensed by pattern recognition receptors (PRRs) of the host induce a strong inflammatory response that manifests itself primarily with fever, even in the absence of bacteremia [100,101]. Fever occurs when pyrogenic factors (exogenous or endogenous toxins) induce various types of cells to release pyrogenic cytokines (e.g., interleukin-6, interleukin-1, tumor necrosis factor and interferon). These cytokines stimulate the hypothalamus to the release of prostaglandin E2, through which the temperature setpoint of the hypothalamus is raised, resulting in an increased body temperature [102]. In conclusion, MBI is a more important determinant of fever than neutropenia, and it is highly associated with infectious complications after intensive cytotoxic therapy and HCT. The concept of ‘febrile neutropenia’ alone no longer suffices, as fever may well be the result of an inflammatory response to MBI, whether or not infection is involved. Therefore, the paradigm ‘febrile mucositis’ proposed by van der Velden et al. [69] reflecting the central role of MBI in the triad of MBI, inflammation and BSI seems more appropriate.

## 6. Treatment of Mucositis-Related Fever

The duration and incidence of fever, antibiotic therapy, parenteral narcotic use, total parenteral nutrition, and the length of stay in a hospital are all correlated with the severity of oral mucositis graded by WHO scale, as is the risk for significant infections and mortality [8]. Spielberger et al. showed that the recombinant human keratinocyte growth factor Palifermin significantly reduced both the incidence of oral mucositis and febrile neutropenia in patients receiving a AHCT after conditioning therapy containing total body irradiation, although Palifermin had no effect on the duration of neutropenia [103]. The use of recombinant human IL-11 preserved the gut barrier after cytotoxic therapy and resulted in a decrease in gut-related BSI [104]. Unfortunately very few drugs have been developed for this purpose or suffered from flaws in their study design. There are several non-randomized and nine randomized controlled trial reports supporting the efficacy of cryotherapy by applying ice chips or ice cold water during high-dose chemotherapy or 15–30 min before and continued until 6 h after the end in preventing grade 3–4 OM in AHCT from 23–85% to 3–33% [105]. In a prospective clinical study, AHCT patients were randomized as to whether or not they were supported with cryotherapy during HDM infusion. Patients supported with cryotherapy experienced a statistically significant lower occurrence of all grades OM, IV opioids and TPN administration, and a reduced need of IV antibiotic administration for febrile episodes. They observed a trend toward a lower incidence rate of FUO episodes (FUO was defined as a temperature >38.3 °C occurring during neutropenia or after neutrophil recovery, with no obvious infectious source or microbiologically documented infections despite appropriate investigation) [106]. IL-1α and IL-1β are potent inflammatory cytokines that activate local and systemic inflammatory processes and are involved in protective immune responses against infections. Research conducted the past few years elucidated in more detail the role of interleukin-1 (IL-1) in the pathogenesis of therapy-induced mucositis [107]. Preclinical animal models on the role of IL-1 and IL-1 inhibition have shown IL-1 to be a potential therapeutic target for chemotherapy-induced mucositis. Research in murine models has shown IL-1 inhibition to be effective in attenuating mucositis induce by a number of different chemotherapeutic regimens [108,109]. We confirmed that the IL-1/CCXL-1 axis was instrumental in the development of melphalan-induced mucositis using our newly developed murine model of melphalan-induced mucositis. Anakinra, compared to the placebo, decreased the severity of mucositis as indicated by the citrulline course, protected against melphalan-induced weight loss, improved food intake, and lowered body temperature [110]. Importantly, other researchers reported earlier that citrulline reliably reflects small bowel enterocyte mass uninfluenced by the extent of inflammation [111]. Furthermore, anakinra largely protected animals from melphalan-induced microbial injury and pathogen expansion [110]. Treatment with anakinra did not affect the degree and duration of neutropenia. Anakinra alone prevented diarrhea and in combination with dexamethasone protected rats from villous atrophy and reduced idarubicin-induced jejunal apoptosis [112]. IL-1 inhibition with the IL-1 receptor antagonist anakinra has shown to be safe in other human diseases, and much clinical experience with anakinra is available, including some experience in hematology and HCT settings [107]. A phase IIa randomized placebo-controlled clinical trial testing anakinra in patients with multiple myeloma, treated with HDM followed by AHCT accomplished accrual of patients recently and analysis of its results on clinical events of fever during neutropenia, inflammation, mucositis, BSI and microbiome changes are eagerly awaited [113]. It is interesting to speculate on the use of anakinra as mucosa-directed therapy, but as in many studies targeting one cytokine or part of a cellular pathway, even when involved in the initiation phase, the question is whether this single-target approach will be sufficient to counteract all or at least sufficient factors involved [114]. 

## 7. Future Directions

To manage patients with mucositis and infection, several strategies can be applied. The most powerful strategy would be to prevent or treat mucositis in hematology patients, but current available therapeutic options are limited. The MASCC/ISOO guideline [5] recommends in specific situations for OM: recombinant human keratinocyte growth factor (KGF-1/palifermin), photobiomodulation (previously termed ‘low-level laser therapy’), patient-controlled analgesia with morphine and oral cryotherapy. For gastrointestinal mucositis, the MASCC/ISSO guideline recommends octreotide. The therapeutic options are mainly directed toward relief of mucositis-induced symptoms. The lack of clinical guidance for gastrointestinal mucositis highlights the need to improve our understanding of the underlying mechanisms and risk factors. The second strategy therefore might be to identify patients at significant risk of mucosal injury and adopt a more risk-adapted approach in supportive measures to control mucositis or mucositis-related complications such as systemic inflammation, fever and infectious complications. The strongest level of evidence supports dosimetric parameters as key predictors of mucositis risk (ref Wardill scc 2020) [115]. Genetic variants in drug-metabolizing pathways, immune signaling, and cell injury/repair mechanisms were also identified to impact mucositis risk. Factors relating to the individual were variably linked to mucositis outcomes, although female sex and smoking status showed some association with mucositis risk. These risk factors can be complemented by grading the risk of bacteremia using citrulline as a biomarker [86] or grading microbial injury and subsequent risk of bacteremia in HCT recipients [71]. The risk of bacteremia increases with the intensity of the conditioning regimen. Both articles provide a grading system of chemotherapeutic regimens that translates the risk of bacteremia. The next strategy would be to provide preventive and therapeutic antimicrobial therapy based on the anticipated risk of translocating bacteria reserve antimicrobial therapy for high-risk regimens and reduce overuse of antimicrobial agents in lower-risk regimens. The benefits in reducing resistance, dysbiosis and secondary post-transplant complications are obvious. It seems logical to incorporate this risk-adapted strategy in the various antimicrobial stewardship programs implemented in many hospitals. The fourth strategy is to shorten antibiotic use in neutropenic patients without microbial infections found for (persistent) fever in the presence of mucositis based on the grading systems. Recent studies in febrile neutropenic patients clearly show that this approach is safe and feasible irrespective of the neutrophil count [116,117,118,119]. Novel strategies to modify the microbiome [54,120,121], the mucus [122], the bile acids [123], the permeability [9]) and immunological defense [124] are very promising, but the results of further testing in clinical trials needs to be awaited before application in the clinic is standardized. Recently, organoid models for studying mucosal barrier injury have been developed [125]. Organoids are complex 3D culture systems, which represent in vitro miniaturized and simplified model systems of organs, such as the intestines. Although becoming increasingly popular, their development is still in its infancy, and currently a number of limitations preclude its optimal use, such as a limited level of maturity and function, and heterogeneity in organoid systems [126]. The microbial metabolome offers new and promising targets to intervene, for example SCFA loss due to bile acid malabsorption in mucositis parallels existing data in other human conditions [127]. A number of other targets and agents have been identified and are being explored: other pro-inflammatory cytokines and chemokines [128], TLR signaling [129], glucagon-like peptide (GLP) signaling [130] and mucosal strengthening [61]. Finally, fecal microbial transplantation to restore a healthy microbiome in treating intestinal mucositis is been tested [131]. However, this development is still in its infancy, and it is associated with a number of risks and challenges [132]. For many potential therapies, the available evidence is difficult to assess because of design flaws, small sample sizes and heterogeneous patient populations and treatment modalities. Well-designed RCTs are therefore required to establish the effect of these therapies. Future developments such as machine-learning may be used to find pre-treatment risk profiles to predict whether a patient will develop mucositis complicated by BSI [133].

## 8. Concluding Remarks

Survival in patients with hematological malignancies has improved over the years, both due to major developments in anticancer treatment, as well as in supportive care. Nevertheless, important and debilitating complications of intensive treatment regimens still frequently occur, including mucositis, fever and bloodstream infections. Exploring potential interacting mechanisms and directed therapies to counteract mucosal barrier injury is of the utmost importance if we are to continue to improve care for this increasingly growing patient population.

## Figures and Tables

**Figure 1 ijms-24-09592-f001:**
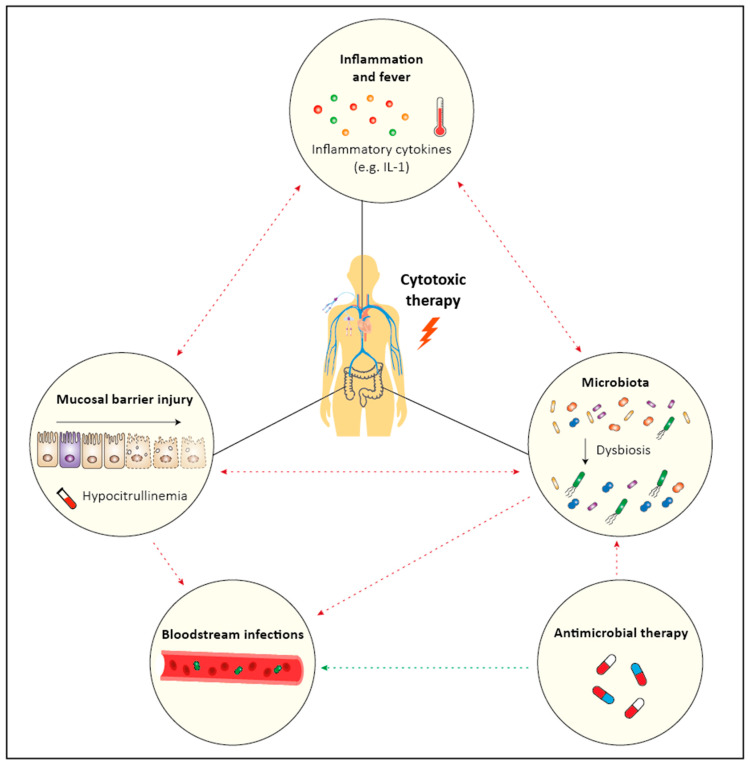
This figure shows the complex interaction between factors involved in inflammatory and infectious complications caused by intensive cytotoxic therapy. Cytotoxic therapy leads to MBI, which is related to inflammation and fever, bloodstream infections and dysbiosis. Although effective in preventing or treating bloodstream infections, antimicrobial therapy is also associated with dysbiosis and the development of antimicrobial resistance (Red Arrow: negative effect, Green Arrow: positive effect). We refer for detailed information to the following reference of Fatizzo and others [73].

## Data Availability

Not applicable.

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
