# Peer review of "Mucositis and Infection in Hematology Patients"

_ijms, 2023, doi:10.3390/ijms24119592_

Round 1
Reviewer 1 Report (Previous Reviewer 2)
In my opinion, the revised manuscript "Mucositis and infection in hematology patients" can be considered for publication.
The content has been extensively revised, with considerations about biological mechanisms and clinical evolutions.
The bibliographic resources have been reviewed, updated, and enriched.
A small recommendation: the abstract reported in the IJMS Platform presents some small discrepancies with respect to that of the Manuscript. In addition, the last sentence is written in first person ("In this perspective I highlight recent advances in our understanding of the relation of mucositis and infection"), while the authors are two. Please, correct.
Author Response
we want to thank the reviewer for the notion that the content has been extensively and succesfully revised.
The reviewer's recommendation about the small discrepancies between the abstract and the manuscript is well taken but had already been changed accordingly when uploading the revised manuscipt. We noticed that the revised manuscript offered to the reviewer contained the original abstract and not the revised one as depicted here above in the Platform.
But we spotted some typo's and therefore upload a corrected abstract and added the word dysbiosis to it.

Reviewer 2 Report (New Reviewer)
It is not only for the hematology patients but also for the cancer patients who received chemotherapy and radiotherapy, the management of oral mucositis in patient is essential.
The perspective article entitled “Mucositis and infection in hematology patients” will help the development of treatment in mucositis in the future. Please see the comment s below
1) In our prospective, two-centre study we investigated the dynamics of micro-bial changes in relation to the development of ulcerative oral mucositis in fifty-one pa-tients with multiple myeloma homogenously treated with high-dose melphalan followed by autologous hematopoietic stem cell transplant (AHCT).[31]
It is better to show the ethics committee Approval Number at your hospital.
2 )Figure 1
Is it shown two factors such as Mucosal barrier injury and Microbiota are important?
Please explain the red and green arrows in Figure 1.
For the Figure 1, the review will help you as below.
Recent insights into the role of the microbiome in malignant and benign hematologic diseases
Critical Reviews in Oncology/Hematology
https://doi.org/10.1016/j.critrevonc.2021.103289
Author Response
we want to thank the reviewer for his comments. We agree with the statement of the reviewer that "it is not only for the hematology patients but also for the cancer patients who received chemotherapy and radiotherapy, the management of oral mucositis in patient is essential".
Comment 1:
In our prospective, two-centre study we investigated the dynamics of micro-bial changes in relation to the development of ulcerative oral mucositis in fifty-one pa-tients with multiple myeloma homogenously treated with high-dose melphalan followed by autologous hematopoietic stem cell transplant (AHCT).[31]. It is better to show the ethics committee Approval Number at your hospital.
Reply:
the number of Ethics committee Approval is NL52117.018.15. we 've added this in the main body of text.
Comment 2 )Figure 1
Is it shown two factors such as Mucosal barrier injury and Microbiota are important
Please explain the red and green arrows in Figure 1.
Reply: we added in the legend the following words
green arrow: positive effect
red arrow: negative effect
comment reviewer:
For the Figure 1, the review will help you as below.
Recent insights into the role of the microbiome in malignant and benign hematologic diseases
Critical Reviews in Oncology/Hematology
https://doi.org/10.1016/j.critrevonc.2021.103289
Reply we added this reference and refer to the review in the body of text
This manuscript is a resubmission of an earlier submission. The following is a list of the peer review reports and author responses from that submission.
Round 1
Reviewer 1 Report
In the present manuscript named’ Mucositis and infection in hematology patients’, Nicole and Charlotte provide an overview of the complications associated with intensive treatment regimens for hematological malignancies, with a particular focus on mucositis and infection. However, there are a few questions need to be addressed before considering for publication.
Comments:
1. The topic seemed nothing attractive as anti-cancer treatment derived mucositis and infection are generally related and the mechanism like immune system damage and unbalanced microbiome populations caused by the treatment is known. Unless the authors could provide other specific reasons to emphasise the importance, this study will not be that interesting and beneficial to the general readers.
2. It is difficult to figure out that the authors really answer their own questions raised in this manuscript in each paragraph.
3. The introduction provides a general overview of the current state of research on mucositis and infection in hematology patients. However, it lacks specific information on the prevalence of these complications among hematological malignancy patients. This information is crucial to contextualize the importance of this topic.
4. The authors have summarized the findings from selected studies, but the results seemed not support their ideas that mucositis and infection came from different mechanisms.
5. While the discussion section provides some insights into potential mechanisms underlying mucosal barrier injury and potential therapies to counteract this injury, however it lacks a critical analysis of the limitations of the studies reviewed and lacks specific recommendations for future research directions, which would help readers understand how to build upon this work.
The authors need to provide a more balanced assessment of the evidence to ensure that readers have a clear understanding of its limitations.
6. The references are comprehensive but do not provide enough context or analysis to support the arguments made in this article.

Reviewer 2 Report
The manuscript "Mucositis and infection in hematology patients" presents interesting aspects of the effects of anticancer treatment in hematology patients.
I suggest the manuscript is acceptable with the following minor revisions:
- The 2nd part of the manuscript contains more citations and seems more detailed than the 1st part. Please, add further references to better illustrate the issue.
- Please, could you add some hypotheses or findings on the biological mechanisms (signaling, factors, etc) underlying the phenomena of mucositis/infections in the specific conditions of the hematological malignancies? If there were few studies, this is an aspect to report.
- Could the conclusions be more directed, suggesting more concrete objectives for both clinical and research?